# MES SV40 Cells Are Sensitive to Lipopolysaccharide, Peptidoglycan, and Poly I:C Expressing IL-36 Cytokines

**DOI:** 10.3390/ijms231911922

**Published:** 2022-10-07

**Authors:** Cesar G. Pelcastre-Rodriguez, Ernesto A. Vazquez-Sanchez, José M. Murrieta-Coxca, Sandra Rodríguez-Martínez, Juan C. Cancino-Diaz, Mario E. Cancino-Diaz

**Affiliations:** 1Departamento de Inmunología y Microbiología, Instituto Politécnico Nacional, Escuela Nacional de Ciencias Biológicas, Mexico City 11340, Mexico; 2Instituto Politécnico Nacional, Escuela Nacional de Medicina y Homeopatía, Mexico City 07320, Mexico; 3Placenta Lab, Department of Obstetrics, University Hospital Jena, 07747 Jena, Germany

**Keywords:** IL-36 cytokines, PAMPs, mesangial cells, inflammation, glomerulus

## Abstract

Mesangial cells (MC) maintain the architecture and cellular communication and indirectly join in the glomerular filtration rate for the correct functioning of the glomerulus. Consequently, these cells are activated constantly in response to changes in the intraglomerular environment due to a metabolic imbalance or infection. IL-36, a member of the IL-1 family, is a cytokine that initiates and maintains inflammation in different tissues in acute and chronic pathologies, including the skin, lungs, and intestines. In the kidney, IL-36 has been described in the development of tubulointerstitial lesions, the production of an inflammatory environment, and is associated with metabolic and mesangioproliferative disorders. The participation of IL-36 in functional dysregulation and the consequent generation of the inflammatory environment by MCs in the presence of microbial stimulation is not yet elucidated. In this work, the MES SV40 cell cultures were stimulated with classical pathogen-associated molecular patterns (PAMPs), mimicking an infection by negative and positive bacteria as well as a viral infection. Lipopolysaccharide (LPS), peptidoglycan (PGN) microbial wall components, and a viral mimic poly I:C were used, and the mRNA and protein expression of the IL-36 members were assessed. We observed a differential and dose-dependent IL-36 mRNA and protein expression under LPS, PGN, and poly I:C stimulation. IL-36β was only found when the cells were treated with LPS, while IL-36α and IL-36γ were favored by PGN and poly I:C stimulation. We suggest that the microbial components participate in the activation of MCs, leading them to the production of IL-36, in which a specific member may participate in the origin and maintenance of inflammation in the glomerular environment that is associated with infections.

## 1. Introduction

Several agents, ranging from bacterial, viral, fungal, and protozoal organisms, trigger renal infections, and the severity depends on factors related to the specific causal agent and the host immune response. Consequently, infections are associated with renal lesions, including acute tubular necrosis, tubulointerstitial nephritis, vasculitis, and glomerulonephritis (GN). Therefore, glomerular injury is mostly the result of immune-mediated mechanisms [1,2].

Mesangial cells maintain the architecture and cellular communication and indirectly intervene in the glomerular filtration rate for the correct functioning of the glomerulus. Consequently, these cells are constantly threatened in response to changes in the intraglomerular environment due to a metabolic imbalance or infections. Different strategies have been developed in the search for the origin, regulation, and control of metabolic disease associated with kidney damage. However, in most cases, kidney damage develops silently and chronically, almost imperceptibly, causing severe conditions. One of the initial steps of this kind of damage is the alteration in the MC functionality, including hypertrophy and proliferation, the deregulation of mesangial matrix synthesis, the induction of inflammatory factors, reactive oxygen species, and pro-fibrotic factors production [3,4].

The IL-36 family is a group of cytokines belonging to the IL-1 superfamily that has inflammatory properties. It includes three active agonist isoforms: IL-36α, IL-36β, and IL-36γ, and a natural antagonist, IL-36Ra. These cytokines share the same receptor complex, IL-36R-IL-1RAcP. Since its renaming, IL-36 has generated great interest in its pivotal activity in physiological and pathological conditions, mainly in chronic inflammatory diseases, including psoriasis and Crohn’s disease [5,6,7,8]. These cytokines act expanding the inflammatory response, participating in the activation of immune cells favoring the antigen presentation, and the production of inflammatory factors [8]. The participation of IL-36 in kidney damage has been poorly investigated, and it has been proposed that IL-36 maintains and increases the production of inflammatory factors by tubular epithelial cells [9]. A pioneering work described the production of IL-36α in tubular epithelial cells, which correlates with an increase in the development of tubulointerstitial lesions (TILs) [10]. Afterwards, it was shown that IL-36α facilitates the activation of the NLRP3 inflammasome and the activation of the IL-23/IL-27 axis in a unilateral ureteral obstruction model in mice and tubular epithelial cell cultures, and also induces an increase in TILs [11]. Furthermore, it was reported that the lack of IL-36R in kidney tissue reduces ischemia-reperfusion acute damage by decreasing inflammatory factors. Additionally, IL-36α levels in the serum of patients with acute kidney injury correlate with the severity of kidney damage [9]. At the glomerular point, IL-36α and IL-23 production has been associated with mesangioproliferative glomerulonephritis, such as IgA nephropathy, which is a common kidney disease that chronically impairs kidney function [12]. All of the above suggests that the IL-36 cytokine system has pivotal functions linked to renal inflammation.

Currently, the main cause of kidney damage is associated with metabolic dysregulation and infections. In this work, we analyzed the expression of the inflammatory mediators by MCs in the presence of microbial components, focusing on the expression of the IL-36 cytokine system. The IL-36 production by MCs in response to those PAMPs may be an initial step in the beginning and maintenance of the inflammation, which is characteristic of the first stages of glomerular damage.

## 2. Results

### 2.1. MES SV40 Cells Express Inflammatory Mediators under Stimulation with PAMPs

MCs were stimulated with different TLR ligands to evaluate if the selected components generate the activation of inflammatory pathways in these cells, as has already been reported with LPS [13]. Importantly, we detected a strong stimulation with PGN in MCs, triggering an increase in IL-1β mRNA of almost 400-fold more, in comparison to the non-stimulated control (NTC, Figure 1B) (*p* < 0.001). LPS induced a moderate but not significant IL-1β mRNA expression (8-fold increases more than the NTC) (Figure 1A) and a small response with poly I:C (Figure 1C) was found. For IL-6, the highest levels were also detected under PGN stimulation, with levels of around 15-fold (Figure 1B, *p* < 0.01), and a similar induction with poly I:C showed about 10-fold, with respect to the NTC (Figure 1C, *p* < 0.05); however, LPS had a poor induction, with almost five increments (Figure 1A). On the other hand, TNFα was mostly induced under a poly I:C stimulation at about 20-fold, with respect to the NTC (Figure 1C, *p <* 0.01), followed by PGN stimulation at 15-fold (Figure 1B, *p* < 0.001). The induction of TNFα by LPS was the lowest (Figure 1A). This indicates that the selected PAMPs can induce the activation of some inflammatory pathways in MCs to different degrees.

### 2.2. LPS Induces Production of IL-36β in MES SV40

Based on the previous results, we looked for the expression and production of IL-36’s members under LPS stimulation also being inflammatory cytokines that could contribute to the creation of an inflammatory environment. With the same stimulation conditions, only an increase in IL-36β mRNA expression was detected, increasing up to 20-fold more than the NTC (Figure 2B, *p* < 0.001), which was not so for IL-36α and IL-36γ (Figure 2A, C). Interestingly, the MCs showed a higher expression of IL-36β mRNA compared to IL-1β’s expression (Figure 1A), which was the most induced inflammatory component by LPS. We corroborated our results obtained from mRNA by WB, where the MCs showed up to a 3-fold higher IL-36β production than the NTC (*p* < 0.01, Figure 2D). The MCs were stimulated using 100 ng of LPS in 2 mL of medium (the amount that induced the production of IL-36β the most, Figure 2D), and IL-36β was examined by means of immunofluorescence (IF) (Figure 2E). We found that the production of IL-36β is confined to the cytoplasm of cells. Therefore, the results suggest that LPS mainly induces the expression of the messenger (even more than IL-1β, IL-6, and TNFα) and a higher production of IL-36β protein in MCs.

### 2.3. IL-36a Is Induced after Stimulation with PGN in MES SV40

We observed that PGN generated a strong inflammatory response in the MCs (Figure 1B), and we wondered if it might also induce an expression increase in the IL-36 members. In contrast to LPS, the PGN stimulation induced mainly IL-36α with almost 40-fold increases in comparison to the NTC (*p* < 0.001) and a dose-response effect was observed (Figure 3A). An increase in IL-36γ mRNA was detected, but no significant differences were found (Figure 3C), while IL-36β presented similar levels to the NTC (Figure 3B). IL-36α induction was higher than IL-36β induction with LPS (Figure 2B), with about a 40-fold increase (*p* < 0.001, Figure 3A). Although PGN caused the increase in IL-36α mRNA to a comparable level to IL-6 and TNFα, the levels of expression of IL-36α were less than the IL-1β induction (Figure 1B). This may be a consequence of synergy between IL-36α and IL-1β. Similarly, the induction of IL-36α was verified by WB, detecting around 3-fold increases of IL-36α protein in comparison to the NTC (*p* < 0.001, Figure 3D). The detection of IL-36α by IF was also performed after stimulation with 1 µg of PGN. We observed that akin to IL-36β, IL-36α is located in the cytoplasm of MCs. It suggests that stimulation with PGN triggers a strong inflammatory response in MCs, inducing IL-36α expression in addition to the expression of IL-1β, IL-6, and TNFα, which could contribute to the maintenance of the inflammatory circuit.

### 2.4. Poly I:C Induces Expression of IL-36α in MES SV40

Poly I:C induced the activation of the inflammatory pathways in MCs in similar conditions to PGN, but the lowest amount of poly I:C (0.1 ng) was enough to induce significant levels of IL-36α mRNA (Figure 4A, *p* < 0.01). This low poly I:C concentration also induced the mRNA expression of the other inflammatory cytokines, IL-1β, IL-6, and TNFα, in these cells (Figure 1C). This suggests that this component achieves faster activation of the MCs than the other analyzed PAMPs. The IL-36β mRNA expression was not detected under any poly I:C amount in the MCs (Figure 4B). By contrast, IL-36γ mRNA expression was slightly observed, with no significant differences with respect to the NTC (Figure 4C, *p* < 0.01). Similar to other PAMPs, we corroborated the PCR results by Western blot. The production of IL-36α was also detected after stimulation with the lowest concentration of poly I:C (Figure 4D, *p* < 0.01). The IL-36α protein was located in the cytoplasm of MCs, as visualized by IF (Figure 4E).

### 2.5. The IL-36 Receptor and the Antagonist mRNA Expression Is Not Induced by PAMPs in MES SV40

None of the selected PAMPs induced any significant change in the mRNA expression of the IL-36 receptor (IL-36R) and IL-36 receptor antagonist (IL-36Ra) in the MES SV40 cells (Figure 5A–C). A slightly significant expression of IL-36R mRNA after the PGN treatment (Figure 5B) and IL-36Ra mRNA after treatment with LPS was observed (Figure 5A).

## 3. Discussion

The IL-36 cytokine system is gaining interest due to its association with the development of renal damage. However, little is known about the triggers of IL-36 expression by renal cells. This work aimed to show the mesangial cell response against the microbial components, including LPS, PGN, and poly I:C (Gram-positive, Gram-negative bacteria, and viral components, respectively) expressing IL-36 cytokine members.

Using the mouse mesangial cell line (MES SV40), we observed that LPS generated cell activation, increasing IL-1β mRNA expression. However, the LPS-induced IL-36 mRNA expression was greater than the IL-1β mRNA expression. This may be one of the initiation factors of glomerular inflammation associated with bacterial infections. Further, our result suggests that in MCs, IL-36β could be one of the initial cytokines produced during an inflammatory process due to the faster expression of IL-36β. The IL-36β expression was induced by LPS after 2 h of stimulation, and minor concentrations were required, while the TNFα, IL-6, and IL-1β expressions were detected after longer time points, and more high concentrations of LPS were needed. Remarkably, despite the fact that the IL-36 isoforms use the same receptor complex, there are important differences in the responses they elicit. In particular, IL-36β has been shown to be a potent inducer of maturation, activation, proliferation, and cytokine production (such as IFNy and IL-2) in immune cells, favoring the development of a strong innate inflammatory response [14,15,16,17]. In addition, unlike to IL-36α and IL-36γ production, (this begins in non-immune cells and subsequently acts in immune cells favoring the inflammation maintenance), IL-36β seems to be restricted to immune cells [5,6,14,18].

It is known that increases in glucose concentrations lead to an escalation of TLR-4 expression and activity in mouse MCs. There is evidence that TLR-4 stimulation in MCs induces the production of a great variety of inflammatory molecules and the activation of complex molecular networks associated with diabetic nephropathy [19]. The MCs stimulated with high glucose concentrations enhanced the production of IL-1β, and the presence of LPS works synergically, increasing IL-1β production [13]. Based on our results, we suggest that the LPS-induced early production of IL-36 may help increase the IL-1β production in MCs.

Another cause of kidney damage is also associated with metabolic disorders and chronic low-grade inflammation [20,21]. It has been reported that a high-fat diet produces abnormalities in the intestinal barrier, increasing its permeability and allowing the passage of microbial components, such as LPS, PGN, and lipoteichoic acid, among others, to the bloodstream [20,22,23]. These PAMPs may be retained in the kidneys after blood glomerular filtration, and such accumulation of microbial components could activate MCs to induce the cytokine production, including IL-36 cytokines. This can contribute to the beginning of chronic low-grade inflammation. Here, we showed that the in vitro MES SV40 cells under LPS stimulation produce IL-36. However, more studies are needed to elucidate whether, in vivo, the PAMP accumulation by the glomerulus and the mesangial cell activation occurs. It is known that LPS is captured in serum by the LPS-binding protein (LBP) facilitating its binding to the TLR4/CD14 complex, favoring its signaling pathway. However, there are reports suggesting that LBP also works as a detoxification mechanism where LPS is captured and retained by LBP and by chylomicrons, avoiding an uncontrolled inflammatory response [24]. Although all the stimulations were under serum-free conditions, we observed an LPS-induced expression of inflammatory cytokines in MES SV40. This suggests that an LBP absence is not a limiting condition for the MCs’ activation and induction of inflammation.

To date, there are many reports involving stimulation with PGN in several cells; however, this work represents the first evidence of MC activity under stimulation with a TLR-2 agonist. Unlike LPS, where the beta isoform of IL-36 was detected, in our assay using PGN, the IL-36α isoform was found to be induced. This suggests a specific MC response according to the type of TLR ligand. The IL-36α mRNA and protein expression presented a dose-response behavior. The PGN-induced IL-36α production could have relevance in the field of bacterial-associated glomerulonephritis. This pathology is common in children and is caused mainly by group A *streptococci*, but it can also be caused by Gram-negative bacteria, viruses, and parasites [25]. Although it is accepted that glomerular lesions are caused mainly by complement fixation, the mechanisms by which glomerular damage is generated in this type of nephritis are not completely known. However, it has been described that some complement molecules do not have a lysis effect on MCs; instead, the complement has a sub-lytic effect, thus promoting MC activation and the induction of apoptosis [12,26]. In this context, in a rat nephritis model induced by the administration of C5b-9 and the anti-Thy-1 antibody, MCs producing IL-36 were found. Here, the production of IL-36α and IL-23 was induced via KLF4 and PCAF [12]. According to our findings, the existence of synergism between PGN and the complement molecules in the activation of the MC favoring the IL-36α expression could be suggested. This may generate an inflammatory environment and the acute response that appears in these types of glomerulonephritis.

Viral infections are also associated with various types of glomerulonephritis, either by a direct or indirect mechanism for damage development [27]. Although the exact mechanism by which glomerular damage develops is still unknown, the presence of TLR-3 in mouse kidneys suggest the contribution of viral infections [28]. TLR-3 is the only known receptor whose signaling depends on the TRIF adapter molecule (Toll-IL-1 receptor domain-containing adapter inducing IFN-β) and the RNA helicase RIG-1, with the consequent induction of IFN-responsive genes. Initially, the TLR-3 expression was restricted to dendritic cells and other cells of the immune system. However, in a murine model of systemic lupus erythematosus, particularly in kidneys, the expression of TLR-3 was also described specifically in MCs and infiltrating antigen-presenting cells [28]. To date, no more studies have been carried out involving the stimulation of MCs with viral components, including those mimics, such as poly I:C, as we studied in this work. However, studies involving other cell types, such as placental cells, had demonstrated an increase in IL-36α, IL-36β, IL-36γ mRNA, and protein expression when the cells were treated with poly I:C. The authors reported that poly I:C had a stronger response than that observed with LPS [29]. In keratinocytes, a stimulation with low doses of poly I:C induces the expression and release of soluble IL-36γ in a dose- and time-dependent manner [30]. Our results show that MCs also respond expressing IL-36α and IL-36γ mRNA in response to low amounts of poly I:C (0.1 ng) but not expressing IL-36β mRNA.It has been reported that it is possible that the IL-36 isoforms may present different kinetics under the same stimulation conditions. In fact, there is evidence that the expression of one IL-36 isoform can induce the expression of the others in a positive feedback mechanism [7].

The IL-36 system in viral infections has been reported to have specific patterns of predominant isoform expression, depending on the infected tissue and the virus. For example, increased IL-36α and IL-36γ but not IL-36β mRNA expression has been reported in bronchial epithelial cells after TLR-3 stimulation [31]. In addition, the infection of mouse skin with HSV-1 increased IL-36α and IL-36β but not IL-36γ production [32]. In contrast, an infection with HSV-2 increased only IL-36γ expression in vaginal epithelial cells, inhibiting the spread of HSV-2 by generating an inflammatory environment [33]. Both studies highlight the participation of the IL-36 system in the early stages of infection. This was also observed in this work, since the lowest concentration of poly I:C generated a rapid response of MCs increasing the IL-36 mRNA synthesis as well as protein production. All of the above suggest that MCs may be the first glomerular defense line against microbial elements, releasing inflammatory factors.

On the other hand, IL-36α and IL-36β production was observed in the cytoplasm of MCs, which could suggest the possibility that IL-36 is released into the extracellular milieu, being able to exert their effects in other glomerular cells. An autocrine regulation could also be suggested, although the IL-36R and IL-36Ra mRNA were not induced under our experimental conditions. The release of IL-36 cytokines may trigger the expansion of the inflammatory response, inducing the production of other inflammatory components and generating an inflammatory loop that increases and maintains the inflammation. This could cause MC disturbance, chronic damage, and a loss of glomerular function. The release of IL-36 can also support NLRP3 inflammasome formation and the activation of the IL-17/IL-23 axis in renal tissue. This induces an increase in the inflammatory and fibrotic factors in tubular epithelial cells that allow the formation of TILs, which is a characteristic feature in chronic kidney disease [11] underlining the importance of IL-36 in the early stages of renal damage.

## 4. Materials and Methods

### 4.1. Cell Culture

The mouse mesangial cell line MES SV40 (American Type Culture Collection, ATCC, Manassas, VI, USA) was cultured in 6-well plates (1.5 × 10^6^ cells) in Dulbecco’s Modified Eagle Medium (DMEM) supplemented with the Nutrient mixture F-12 medium (both from Thermo Fischer Scientific, Waltham, MA, USA) in a 1:1 ratio, containing 5% fetal bovine serum (FBS) (Gibco, Waltham, MA, USA) at 37 °C, in a humid atmosphere of 5% CO_2_/ambient oxygen. Both media contain < 10 pg/mL of endotoxin, according to the information of the provider. 

### 4.2. Cell Stimulation

Before the PAMP stimulation and when the cells reached 95% confluence (1.5 × 10^6^ cells/well), the medium was changed by DMEM-F-12 (1:1) without FBS per 1 h. The non-stimulated control groups were maintained in similar conditions throughout all experiments. The cells were stimulated in a total volume of 2 mL of medium with different amounts of the following microbial components: 0.1, 1, 10, 100, 10^3^, and 10^4^ ng to lipopolysaccharide (Escherichia coli´s LPS, Sigma-Aldrich, St. Lois, MO, USA Catalog L2630) for 2 h; 0.1, 1, 10, 100, and 1000 ng to poly (I:C) (polyinosinic-polycytidylic acid, Sigma-Aldrich, St. Lois, MO, USA Catalog P1530) for 2 h, and the same 0.1, 1, 10, 100, and 1000 ng to peptidoglycan (Staphylococcus aureus PGN, Sigma-Aldrich, St. Lois, MO, USA Catalog 77140) for 6 h.

### 4.3. mRNA Measurement by RT-PCR

Once we finished the time stimulation, the medium was removed, and the cells were harvested for total RNA extraction using the TRIzol reagent (Invitrogen, Waltham, MA, USA) according to the manufacturer’s instructions. A total of 2 μg of total mRNA were transcribed to cDNA using M-MLV reverse transcriptase (Invitrogen, Waltham, MA, USA Catalog 28025-013) reagents. Afterwards, a semi-quantitative PCR was performed by kits (Meridian Bioscience, Cincinnati, OH, USA Catalog Bio-25044) using specific primers, as listed in Table 1. The IL-1β, IL-6, TNFα, and IL-36 expressions were normalized against Actb as the endogenous gene. The fold change was calculated in comparison to the respective non-stimulated control.

### 4.4. Western Blot Analysis

Total proteins were extracted after each stimulation condition using lysis buffer (Cell Signaling TECHNOLOGY, Danvers, MA, USA Catalog 9803) with protease inhibitors (Sigma-Aldrich, St. Lois, MO, USA Catalog S8830). Proteins were quantified by the DC Protein Assay (Bio-Rad, Hercules, CA, USA Catalog 5000112) following the manufacturer’s instructions. A total of 50 µg of total protein were loaded and run by SDS-PAGE (sodium dodecyl sulfate-polyacrylamide gel electrophoresis). The proteins were transferred to nitrocellulose membranes (Bio-Rad, Hercules, CA, USA Catalog 1620115). The membranes were blocked with 0.2% gelatin for 1 h at RT. Then, the membranes were incubated with the primary antibodies mouse anti-IL-36α/IL-1F6 1:500 (R&D SYSTEMS, Minneapolis, USA Catalog AF2297) and mouse IL-36β/IL-F8 antibodies 1:500 (R&D SYSTEMS, Minneapolis, USA Catalog AF2298), respectively, and were incubated overnight at 4°. The membranes were incubated with the secondary anti-goat antibody, IgG-HRP 1:2000 (Sigma-Aldrich, St. Lois, MO, USA Catalog A8919), for 1 h. Β-Actin was detected using anti-Actin I-19 HRP (Santa Cruz Biotechnology, TX, USA Catalog sc-1616). The blots were developed/visualized with the ChemiDocTM Touch Imaging System (Bio-Rad, Hercules, CA, USA) equipment, and data were analyzed using the Image LabTM 6.0.1 software (Bio-Rad, Hercules, CA, USA). The number of pixels of blot bands was extracted as a measure of protein levels compared to the β-actine loading control.

### 4.5. Immunofluorescence

For immunostaining, 1.5 × 10^6^ MES SV40 cells/well were cultured and stimulated with 100 ng of LPS, 1 ug of PGN or 0.1 ng of poly I:C, respectively, in a 6-well plate for 6 h. To block protein excretion, Brefeldin A (Thermo Scientific, Waltham, MA, USA Catalog: 00-4506-51) was added before each stimulus to each well, according to the manufacturer’s instructions. Once the stimulation was finished, the medium and the stimulus were removed, and the cells were washed with phosphate-buffered saline (PBS) by shaking. Then, the cells were fixed with a 4% paraformaldehyde solution (Sigma-Aldrich, St. Lois, MO, USA Catalog: 158127) for 20 min at RT. Later, a solution of 0.1% sodium dodecyl sulfate (SDS, Thermo Scientific, Waltham, MA, USA Catalog: L3771) was added for 5 min. Afterwards, the SDS solution was removed, and 0.05% Triton ×100 (Sigma-Aldrich, St. Lois, MO, USA Catalog: X100) was added for 5 min to permeabilization. The Triton ×100 solution was removed, and the blocking solution was added (3% albumin) for 30 min. Subsequently, the blocking solution was removed, 3 washes were performed with PBS, and the primary antibodies diluted 1:100 mouse anti-IL-36α/IL-1F6 (R&D SYSTEMS, Minneapolis, USA Catalog AF2297) or mouse anti IL-36β/IL-F8 (R&D SYSTEMS, Minneapolis, USA Catalog AF2298) were added. The slides were incubated overnight at 4 °C. The next day, the primary antibodies were removed, and 3 washes were performed with PBS. Secondary anti-goat IgG antibody FITC-coupled (Invitrogen, Catalog: A16137) was added, and the slides were incubated at RT for 90 min in darkness. Afterwards, the secondary antibodies were removed, and 4 washes with PBS were performed. Finally, a solution of 4’,6-Diamidino-2-Phenylindole, Dihydrochloride (DAPI, Thermo Scientific, Waltham, MA, USA Catalog D1306) was added at a concentration of 1 ug/mL, and the staining was incubated for 20 min at RT in darkness. Then, 4 washes with PBS were performed, and the mounting was carried out using VECTASHIELD^®^ Antifade Mounting Medium (Vector Laboratories, Newwark, CA, USA Catalog: H-1000-10). The fluorescence was visualized using an LSM 710 NLO (Carl Zeiss Jena, Germany) confocal system coupled to an inverted microscope.

### 4.6. Statistical Analysis

All experiments were independently repeated at least 3 times, and data are presented as the mean ± standard error of the mean (SEM). The statistical analysis was done with GraphPad Prism V.8.0.2 software (GraphPad Software, Inc.). A one-way analysis of variance (ANOVA), followed by Dunnett’s analysis for multiple comparisons, was used. A value of *p* < 0.05 was considered statistically significant, as indicated in every figure.

## 5. Conclusions

Taken together, we demonstrated a significative differential IL-36 expression in the MES SV40 cells exposed to LPS, PGN, and poly I:C. We suggest that IL-36 cytokines could be one of the initial factors in the development of glomerular inflammation. However, further investigations are needed to elucidate the involvement of IL-36 cytokines, MCs, and other renal cells as a determining factor in the development of glomerular damage.

## Figures and Tables

**Figure 1 ijms-23-11922-f001:**
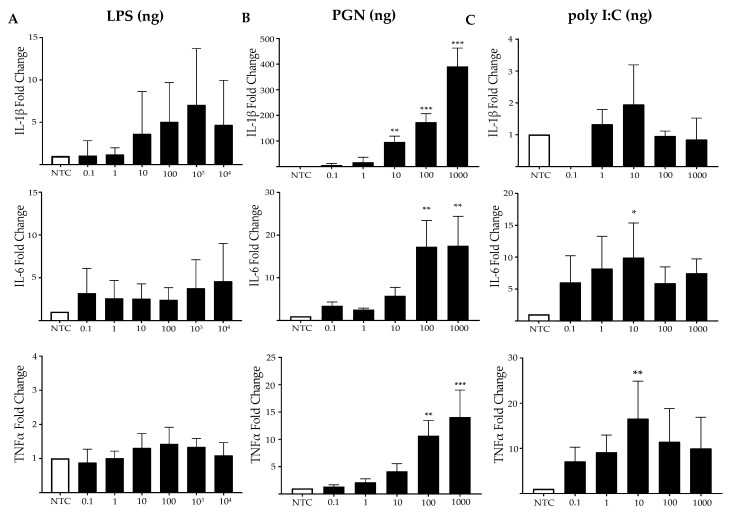
**PAMP stimulation exerts inflammatory response in MCs**. MES SV40 cells (1.5 × 10^6^) were stimulated in 2 mL of medium containing different amounts of (**A**) LPS; (**B**) PGN; (**C**) poly I:C for 6 h, and the mRNA expressions of IL-1β, IL-6 and TNFα were measured. RT-PCR results are expressed as a fold change relative to Actb, and the respective non-stimulated control (NTC). Results are from at least 3 independent experiments and are listed as mean ± SD. * *p* < 0.05, ** *p* < 0.01, *** *p* < 0.001, one-way ANOVA followed by Dunnett´s analysis.

**Figure 2 ijms-23-11922-f002:**
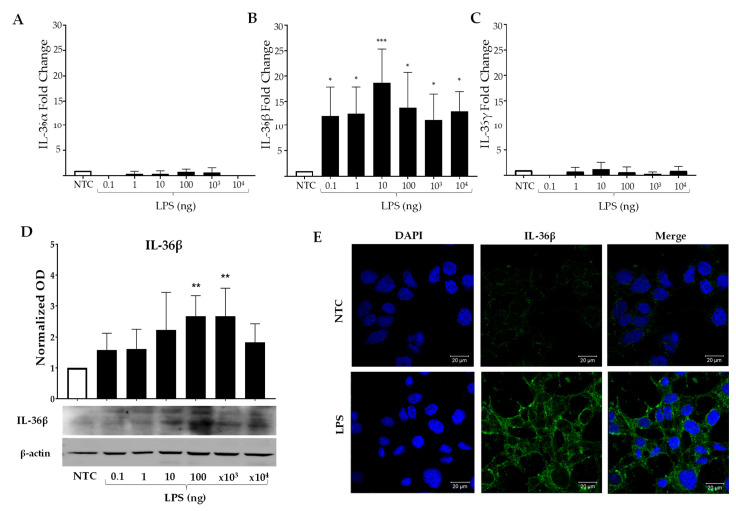
**LPS stimulation triggers a differential expression of IL-36β in MCs.** MES SV40 cells (1.5 × 10^6^) were stimulated in 2 mL of medium containing different amounts of LPS for 2 h, and the expression of (**A**) IL-36α; (**B**) IL-36β; (**C**) IL-36γ mRNA was measured. The expression was normalized against Actb, and the fold change was calculated in comparison with the respective non-stimulated control (NTC). (**D**) Mesangial cells were stimulated for 6 h, and total protein was extracted for Western blot performance. The intensity of the bands was quantified, normalized against β-actin, and the fold change was calculated. E) Representative image of IL-36β production (stained green) in MES SV40 cells stimulated with 100 ng of LPS. The results were obtained from at least 3 independent experiments and are listed as mean ± SD. Data were analyzed with one-way ANOVA followed by Dunnett´s analysis. * *p* < 0.05, ** *p* < 0.01, *** *p* < 0.001.

**Figure 3 ijms-23-11922-f003:**
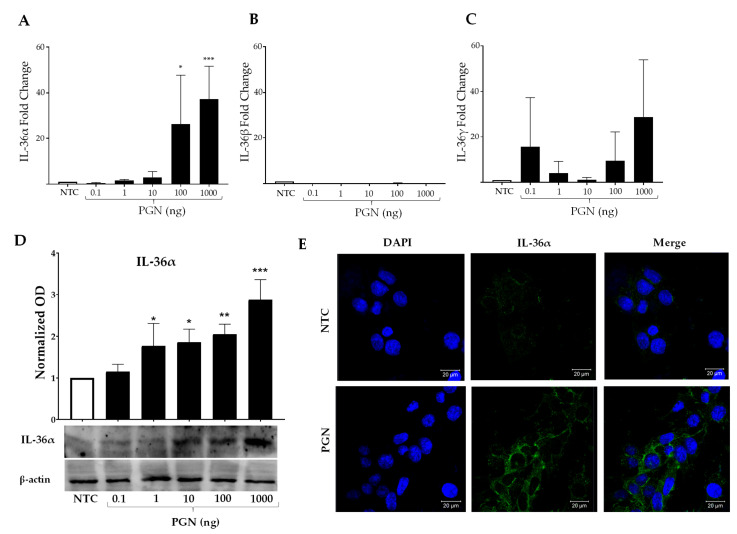
**The stimulation of MCs with PGN generates the production of IL-36α**. MES SV40 cells (1.5 × 10^6^) were stimulated in 2 mL of medium containing different amounts of PGN for 6 h, and (**A**) the IL-36α; (**B**) IL-36β; (**C**) IL-36γ mRNA expressions were measured. The expression was normalized against Actb, and the fold change was calculated in comparison with the respective non-stimulated control (NTC). (**D**) Mesangial cells were stimulated under similar conditions, and total proteins were extracted for WB. The intensity of bands was normalized against β-actin, and the fold change was calculated. E) Representative IF image of IL-36α (green stained) in MCs stimulated with 1 μg of PGN for 6 h. Results were obtained from at least 3 independent experiments and are presented as mean ± SD. Data were analyzed with one-way ANOVA followed by Dunnett´s analysis. * *p* < 0.05, ** *p* < 0.01, *** *p* < 0.001.

**Figure 4 ijms-23-11922-f004:**
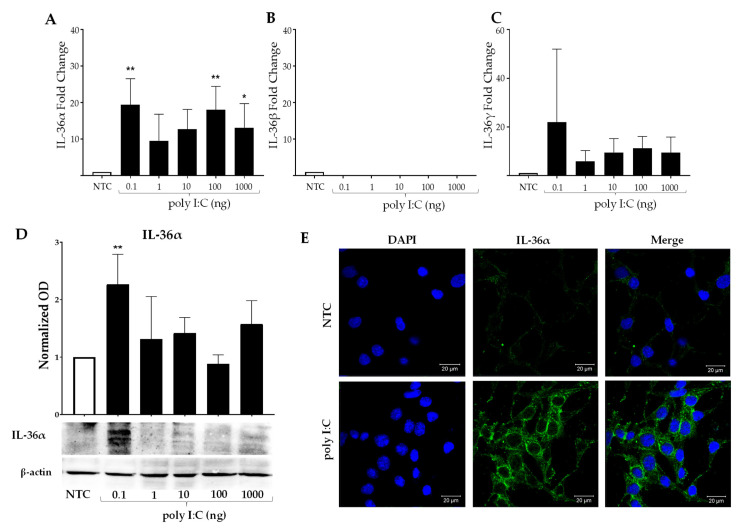
**Poly I:C promotes IL-36α expression in MCs**. MES SV40 cells (1.5 × 10^6^) were stimulated in 2 mL of medium containing different amounts of poly I:C for 2 h, and the expressions of (**A**) IL-36α; (**B**) IL-36β; (**C**) IL-36γ were measured. The expression was normalized against Actb, and the fold change was calculated against the non-stimulated control (NTC). (**D**) For the WB analysis, MCs were stimulated for 6 h under similar conditions, and total proteins were extracted. The intensity of bands was normalized against β-actin, and the fold change was calculated. (**E**) Representative IF image of IL-36α (green stained) in MCs stimulated with 0.1 ng of poly I:C for 6 h. The results were obtained from at least 3 independent experiments and are presented as mean ± SD. Data were analyzed with one-way ANOVA followed by Dunnett´s analysis. * *p* < 0.05, ** *p* < 0.01.

**Figure 5 ijms-23-11922-f005:**
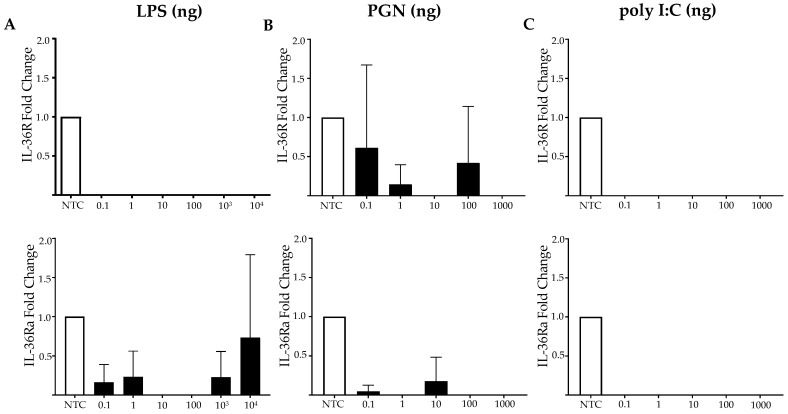
**IL-36R and IL-36RA expressions in MCs after stimulation with PAMPs**. MES SV40 cells (1.5 × 10^6^) were stimulated in 2 mL of medium containing different amounts of (**A**) LPS; (**B**) PGN; (**C**) poly I:C, and the mRNA expressions of IL-36R and IL-36Ra were measured. The expression was normalized against Actb, and the fold change was calculated in comparison to the respective non-stimulated control. The results were obtained from at least 3 independent experiments and are listed as mean ± SD.

**Table 1 ijms-23-11922-t001:** Primer sequences used for semi-quantitative RT-PCR.

Primer (Mouse)	Forward	Reverse
**Actb**	*ATGTGGATCAGCAAGCAGGA*	*AAAGGGTGTAAAACGCAGCTC*
**TNFα**	*CTACCCCCAATGTGTCCGTC*	*GCCGTATTCATTGTCATACCAGG*
**IL-1β**	*TGAAGAAGAGCCCATCCTCTGT*	*GGGTGTGCCGTCTTTCATTAC*
**IL-6**	*CCTCTCTGCAAGAGACTTCCATC*	*AGCCTCCGACTTGTGAAGTGGT*
**IL-36α**	*GCAAACAGTTCCAGTCACTAT*	*GGGTGTCTTTGATTGCTTCTT*
**IL-36β**	*TGCATGGATCCTCACAATC*	*GGCTATAAACCAGCCAGGATA*
**IL-36γ**	*CACAGAGTAACCCCAGTCAG*	*TTGGTCCTGCTTACCTTTCA*
**IL-36R**	*GCAGCAGATACGTGTGAGGAC*	*TTGGTAGCAGTTGTGGGCATT*
**IL-36Ra**	*GGGCACTATGCTTCCGAATG*	*CTTTGATTCCTGGCCCCCGA*

## Data Availability

Not applicable.

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
