# Peer review of "MES SV40 Cells Are Sensitive to Lipopolysaccharide, Peptidoglycan, and Poly I:C Expressing IL-36 Cytokines"

_ijms, 2022, doi:10.3390/ijms231911922_

Round 1

Reviewer 1 Report

The authors of the article obtained interesting results. However, I have a number of comments.

The main observation.

1) The authors used the introduction of LPS amounts from 0.1 ng or higher into the MC culture, probably in a medium volume of 0.2-0.3 ml (LPS concentrations in the medium are not specified by the authors, and this can mislead readers). Meanwhile, in healthy humans, LPS levels in the blood usually do not exceed 10 pg/ml or 0.01 ng/ml (https://doi.org/10.1086/315093).  In metabolic endotoxemia, this level may exceed 2-3 times, but usually does not reach 0.1 ng/ml (doi: 10.3389/fimmu.2020.594150). In sepsis, median LPS values may be 0.3 ng/mL (https://doi.org/10.1086/315093), In Crohn's disease, LPS levels usually do not exceed 1 ng/mL  (https://doi.org/10.1155/2015/843089). 

The authors further attribute the hyperproduction of IL-36 only to chronic inflammation. However, IL-36 can be actively involved in the pathogenesis of sepsis (https://doi.org/10.1093/infdis/jiw535) and acute kidney injury (DOI: 10.1016/j.kint.2017.09.017).

Thus, the LPS phenomena (dose and incubation time) in the peer-reviewed article cannot be viewed as model a low-grade of chronic inflammation. To no lesser extent, the results obtained by the authors may be relevant to acute infectious kidney injury. The same can be said of the other PAMPs used by the authors. Overall, I do not see a scientific rationale for the PAMP doses used by the authors.

Other comments.

2. the authors do not point out that LPS on the TLR4/CD14 receptor complex usually act as part of a complex with blood plasma proteins (LBP, lipoproteins, others).

3. The article should use the same abbreviations IL-36α, IL-36β and IL-36γ, IL-1β (not IL-1b), TNF-α (not Thf), IFNγ (not IFNg).

4. Present results as mean ± standard error only when the distribution is normal. The normality of the distribution was not tested in the article.

5. The Abbreviations section needs to be redone.

Author Response

We appreciate the comments and suggestions of the Reviewers. Two versions of the revised manuscript have been prepared and submitted for revaluation: one version with all modifications highlighted in yellow, and the other version is final. Here, in this letter, the Reviewers’ questions are written in normal characters and our answers in italics.

Revisor 1

The authors of the article obtained interesting results. However, I have a number of comments.

The main observation.

  1. The authors used the introduction of LPS amounts from 0.1 ng or higher into the MC culture, probably in a medium volume of 0.2-0.3 ml (LPS concentrations in the medium are not specified by the authors, and this can mislead readers). Meanwhile, in healthy humans, LPS levels in the blood usually do not exceed 10 pg/ml or 0.01 ng/ml (https://doi.org/10.1086/315093). In metabolic endotoxemia, this level may exceed 2-3 times, but usually does not reach 0.1 ng/ml (doi: 10.3389/fimmu.2020.594150). In sepsis, median LPS values may be 0.3 ng/mL (https://doi.org/10.1086/315093), In Crohn's disease, LPS levels usually do not exceed 1 ng/mL (https://doi.org/10.1155/2015/843089).

The authors further attribute the hyperproduction of IL-36 only to chronic inflammation. However, IL-36 can be actively involved in the pathogenesis of sepsis (https://doi.org/10.1093/infdis/jiw535) and acute kidney injury (DOI: 10.1016/j.kint.2017.09.017).

Thus, the LPS phenomena (dose and incubation time) in the peer-reviewed article cannot be viewed as model a low-grade of chronic inflammation. To no lesser extent, the results obtained by the authors may be relevant to acute infectious kidney injury. The same can be said of the other PAMPs used by the authors. Overall, I do not see a scientific rationale for the PAMP doses used by the authors.

We appreciate the reviewer´s comments and suggestions. We apologize for the confusion regarding the presented data. We stimulated the MES SV40 cells using 0.1, 1.0, 10, 100, 103, and 104 ng of LPS and of the other PAMPs in a final volume of 2 mL of non-supplemented culture medium in a 6-well plate. This was based on already published data for the stimulation with LPS (doi: 10.1055/s-0042-105641). The DMEM (GIBCO Cat. No. 11995) and F-12 (GIBCO Cat. No. 11765) culture mediums used for MES SV40 culture contain < 0.01 EU/mL of endotoxin, which is equivalent to < 10 pg/mL. In addition, we observed very low levels of the inflammatory cytokines in not treated controls which was the medium alone and PBS as the vehicle supporting the effect of the exogenous LPS. This information is shown in the figure’s footnotes and in the material and methods section.

Lines 313-314: Both mediums contain < 10 pg/mL of endotoxin according to the information of the provider.

Lines 319-324: Cells were stimulated in the total volume of 2 mL of medium with different amounts of the following microbial components: 0.1, 1, 10, 100, 103, and 104 ng to lipopolysaccharide (Escherichia coli´s LPS, Sigma Aldrich Catalog L2630) form 2 hours. 0.1, 1, 10, 100, and 1000 ng to poly (I: C) (polyinosinic-polycytidylic acid, Sigma Aldrich Catalog P1530) for 2 hours and same 0.1, 1, 10, 100, and 1000 ng to peptidoglycan (Staphylococcus aureus PGN, Sigma Aldrich Catalog #77140) for 6 hours.

We removed the low-grade inflammation part of the introduction, which is briefly mentioned in the discussion section. The manuscript focus was directed on renal infections by gram-negative and gram-positive bacteria and by viral infections.

Lines 195-199: The IL-36 cytokine system is gaining focus due to its association with the development of renal damage. However, little is known about the triggers of IL-36 expression by renal cells. This work aimed to show the mesangial cell response against microbial components including LPS, PGN, and poly I:C (gram-positive, gram-negative bacteria, and viral components respectively) expressing IL-36 cytokine members.

 Lines 224-241: Another cause of kidney damage is also associated with metabolic disorders and chronic low-grade inflammation [20, 21]. It has been reported that a high-fat diet produces abnormalities in the intestinal barrier, increasing its permeability and allowing the passage of microbial components such as LPS, PGN, and lipoteichoic acid, among others, to the bloodstream [20, 22, 23]. These PAMPs may be retained in the kidneys after blood glomerular filtration and such accumulation of microbial components could activate MCs inducing cytokine production including the IL-36 members. This can contribute to the beginning of chronic low-grade inflammation. Here, we showed that in vitro MES SV40 cells under LPS stimulation produce IL-36.  However, more studies are needed to elucidate whether in vivo the PAMP accumulation by the glomerulus and the mesangial cell activation occurs. It is known that LPS is captured in serum by the LPS-binding protein (LBP) to activate the TLR4/CD14 complex favoring its signaling pathway. However, there are reports suggesting that LBP works as a detoxification mechanism where LPS is captured and retained by LBP and by chylomicrons avoiding an uncontrolled inflammatory response [24]. Although all the stimulations were under serum-free conditions, we observed an LPS-induced expression of inflammatory cytokines in MES SV40. This suggests that LBP absence is not a limiting condition for the MC activation and the induction of inflammation. 

Other comments.

  1. The authors do not point out that LPS on the TLR4/CD14 receptor complex usually act as part of a complex with blood plasma proteins (LBP, lipoproteins, others).

We appreciate the Revisor´s comment. This point was encompassed in the discussion section.

Lines 234-241: It is known that LPS is captured in serum by the LPS-binding protein (LBP) to activate the TLR4/CD14 complex favoring its signaling pathway. However, there are reports suggesting that LBP works as a detoxification mechanism where LPS is captured and retained by LBP and by chylomicrons avoiding an uncontrolled inflammatory response [24]. Although all the stimulations were under serum-free conditions, we observed an LPS-induced expression of inflammatory cytokines in MES SV40. This suggests that LBP absence is not a limiting condition for the MC activation and the induction of inflammation. 

  1. The article should use the same abbreviations IL-36α, IL-36β and IL-36γ, IL-1β (not IL-1b), TNF-α (not Thf), IFNγ (not IFNg).

The abbreviations were homogenized along the manuscript.

  1. Present results as mean ± standard error only when the distribution is normal. The normality of the distribution was not tested in the article.

We changed all presented data as mean ± SD to describe better the data variability. If requested, we can add to supplementary material the QQ graphs from a Shapiro-Wilk test that we applied to our data. The test did not show significant differences suggesting a behavior of a normal distribution.

  1. The Abbreviations section needs to be redone.

The section was updated accordingly

Reviewer 2 Report

An interesting article on mesangial cells, which are sensitive to PAMPs expressing IL-36 cytokines, but the results are mainly based on in vitro MES SV40 mesangial cell line.

Some suggestions:

1. Any justification that why only a mouse MES SV40 was chosen in the study? Any human cell lines or primary animal cells? The authors may consider revising their title to mouse MES SV40 cells. 

2. There are quite a lot of PAMP ligands, what is the reason that only LPS, PGN, and poly I:C were chosen?

3. Has the authors studied the signaling transduction mechanisms of the IL-36a expressions? The finding seems to be in a too preliminary stage and no animal data support.

Author Response

We appreciate the comments and suggestions of the Reviewers. Two versions of the revised manuscript have been prepared and submitted for revaluation: one version with all modifications highlighted in yellow, and the other version is final. Here, in this letter, the Reviewers’ questions are written in normal characters and our answers in italics.

Revisor 2

An interesting article on mesangial cells, which are sensitive to PAMPs expressing IL-36 cytokines, but the results are mainly based on in vitro MES SV40 mesangial cell line.

Some suggestions:

  1. Any justification that why only a mouse MES SV40 was chosen in the study? Any human cell lines or primary animal cells? The authors may consider revising their title to mouse MES SV40 cells.

The title of the manuscript was modified as follows: MES SV40 cells are sensitive to lipopolysaccharide, peptidoglycan, and poly I:C expressing IL-36 cytokines

The mouse mesangial cell line MES SV40 is one of the most stable cell lines available for in vitro studies (doi: 10.14670/HH-15.403). We understand the in vitro limitations, however, our results are congruent with some studies in vivo (doi: 10.14670/HH-15.403; doi: 10.2147/DDDT.S274256).  To our knowledge, there is any available mesangial cell line of human origin and to date, only one source of primary human renal mesangial cells (HRMC) is offered for research (doi: 10.7150/ijbs.20485; doi: 10.1155/2013/754946; doi: 10.1155/2016/7932765).  Unfortunately, we do not have the laboratory conditions to isolate primary material, which needs further standardization, and maintenance and may be contaminated with other cell types. We consider that our results open more questions about mesangial cell biology which may encourage future studies by us and the interested researchers in the field.  

2. There are quite a lot of PAMP ligands, what is the reason that only LPS, PGN, and poly I:C were chosen?

The title of the manuscript was modified mentioning the specific PAMP: MES SV40 cells are sensitive to lipopolysaccharide, peptidoglycan, and poly I:C expressing IL-36 cytokines

To date, only the expression of TRL-4, TRL-2, and TLR-3 as the receptors for LPS, PGN, and poly I:C respectively, has been demonstrated in mesangial cells of mouse and human origin (doi: 10.1681/ASN.2008050549). Therefore, we considered it appropriate to start up with the mentioned compounds.

3. Has the authors studied the signaling transduction mechanisms of the IL-36a expressions? The finding seems to be in a too preliminary stage and no animal data support.

In this work, we did not focus on signaling pathways. The classic LPS, PGN, and poly I:C induced pathways are widely demonstrated (doi.org/10.1016/j.smim.2003.10.003). It is suggested that IL-36 isoforms can induce specific signaling pathways, which may depend on the specific cell type (doi: 10.1016/j.cellsig.2020.109773). We aim to show the induction of IL-36 expression by the aforementioned microbial components in mesangial cells as first evidence and the potential involvement of this cytokine system with the dysregulated mesangial activity and the possible relation with the low-grade inflammation associated with renal malfunction. We understand that further evidence in vitro and in vivo is needed but we consider that our results open more questions about mesangial cell biology, which may encourage future studies by us and the interested researchers in the field.

Round 2

Reviewer 1 Report

The authors took into account all the main comments and suggestions. In this form, the article can be published in IJMS.

Reviewer 2 Report

I have no further comment and the manuscript can be accepted in the revised version.